# Pre-Shaped Burst-Mode Hybrid MOPA Laser System at 10 kHz Pulse Frequency

**DOI:** 10.3390/s23020834

**Published:** 2023-01-11

**Authors:** Shanchun Zhang, Xin Yu, Jiangbo Peng, Zhen Cao

**Affiliations:** National Key Laboratory of Science and Technology on Tunable Laser, Harbin Institute of Technology, Harbin 150001, China

**Keywords:** burst mode, pre-shaped, narrow linewidth, uniform distribution

## Abstract

A temporal pre-shaped burst-mode hybrid fiber-bulk laser system was illustrated at a 10 kHz rate with a narrow spectral linewidth. A theoretical model was proposed to counteract the temporal profile distortion and compensate for the desired one, based on reverse process of amplification. For uniformly modulated injection, amplified shapes were recorded and investigated in series for their varied pulse duration, envelope width and amplification delay, respectively. The pre-shaped output effectively realized a uniform distribution on a time scale for both the burst envelope and pulse shape under the action of the established theoretical method. Compared with previous amplification delay methods, this model possesses the capacity to extend itself for applications in burst-mode shaping with variable parameters and characteristics. The maximum pulse energy was enlarged up to 9.68 mJ, 8.94 mJ and 6.57 mJ with a 300 ns pulse duration over envelope widths of 2 ms to 4 ms. Moreover, the time-averaged spectral bandwidths were measured and characterized with Lonrentz fits of 68.3 MHz, 67.2 MHz and 67.7 MHz when the pulse duration varied from 100 ns to 300 ns.

## 1. Introduction

In recent times, laser-based optical measuring techniques have greatly boosted parameter characterization of physical and chemical processes on the spatial and temporal scales [1,2,3,4,5,6]. For gas-phase flows, the captured imaging intensity is typically provided at a relatively lower level, so a high-energy pulsed source is therefore essential to enhance the signal-to-noise ratio (SNR), especially in individual-shot optical measurements. Turbulent flows feature inherently nonlinear and unstable behaviors, resulting in dynamic variations spanning several orders of magnitude on the scale of temporal distribution [7,8,9]. In this case, highly time-resolved measurements determine that a laser source with a high pulse repetition frequency (PRF) is is required for the acquisition of high-rate images and in order to freeze dynamic variation and track transient processes in turbulent flows. Therefore, measurement coinciding with a high-spatiotemporal-resolution is desired and imperative when characterizing reacting flames and non-reacting jets in turbulent flows. While continuously operated diode-pumped, solid-state (DPSS) lasers can generate pulsed lasers with high frequency by means of Q-switching techniques [10,11], such output is further restricted due to the excessive thermal accumulation occurring inside the gain medium as the injected pump power increases, thus limitations advances in optical diagnosis. The emerging burst-mode method has the capacity to generate high-rate and high-energy pulsed lasers at a lower duty cycle for effective thermal relaxation by grouping a series of sequential pulses into the envelope over a limited duration. This approach corresponds to method for measuring burst-mode laser-based high-rate field parameters illustrated in [12,13,14,15]. 

Numerous optical-based imaging techniques have been introduced in recent years, with an impressive list consisting of, but not limited to, planar laser-induced fluorescence (PLIF) [16,17,18], laser-induced fluorescence (LIF) [19,20], particle-imaging velocimetry (PIV) [21,22,23], and planar Doppler velocimetry (PDV) [24,25]. Nevertheless, many of them are consistently susceptible and limited to both required active seeding and background noise interference in actual flow fields diagnosis. By comparison, filtered Rayleigh scattering (FRS) acts as a promising method for measuring parameters (e.g., temperature, velocity, pressure, and mixture fraction) due to its strong suppression of scattering noise (Mie scattering and windows) with an identical spectral profile as the source. It also lessens complexity in harsh environments without tracing particles or molecules. Conversely, some additional features of the adopted laser source are required for available FRS optical diagnosis systems with narrow spectral bandwidths and stretched pulse durations. Concretely, such narrow-linewidth sources enable one to purify the transmission of the targeted stimulated Rayleigh–Brillouin scattering (RBS) spectrum with a temperature-induced broadened spectral profile. Moreover, the FRS signal level increases linearly with instantaneous laser energy instead of relying on the peak power associated with pulse duration. This allows for the use of temporally stretched pulses, eliminating the optical breakdown that interfered with detected flows during higher-energy deposition and boosting the signal of energy-based technique. In addition, another vital factor, the burst envelope distribution, could play a significant role in FRS optical imaging because an inhomogeneous burst envelope can have a negative effect on the intensity of consecutively captured time-relevant images, with direct relevance for the characterization of flow parameters. For example, an imhomogenous burst envelope distribution led to a large fluctuation in 1D FRS signal profiles with energy normalization in measuring conditions of ambient air at 295 K utilizing five different shots, as illustrated in [26]. Such variation in results could stem from the thermal effects of the adopted source, resulting in defocusing and non-uniform scattering signals being captured by the photodiodes in the FRS normalization process. Hence, a uniform burst envelope distribution is an essential source requirement to ensure less deviation is induced by the instrument itself. In addition, enhancing measurement accuracy could hold great potential in the assessment and validation of reacting and non-reacting flow models.

For the developments of a pulsed laser amplifier system with a flat-hat profile, a few relevant reports have emerged in recent years. In 2012, Zhu et al. illustrated a single-frequency pulsed laser source with a flat-top profile distribution at 100 Hz repetition rate, realizing a pulse energy of 10 mJ for a 110 ns pulse duration [27]. In 2013, Huang et al. reported a narrow linewidth pulsed laser operated at a 10 Hz rate with a 5 μs flat-top macro-pulse distribution [28]. In 2014, Slipchenko et al. exhibited a 100 kHz narrow linewidth, burst-mode laser with dual-wavelength diode-pumped amplifiers and which incorporated an arbitrary waveform generator to provide uniform burst profile shapes [29]. Furthermore, Xie et al. demonstrated a temporal envelope system with programmable burst-mode all-fiber amplifiers in 2022, yielding a flat-top burst profile distribution of a pre-shaped output [30]. Based on previous reports, generating flat-hat profiles has mainly been the focus of experimental methods with programmable inputs, which bring about some inconvenience in uniformly shaping pulsed laser amplifier systems with variable parameters. Concerning burst-mode sources, the envelope distribution, within a burst of pulses, is subject to several factors involved in envelope duration, input energy, pump peak power, pulse interval, and amplification delay (pump time before envelope), respectively. Typically, a distribution trend is found with the obvious deficiency of a lower gain fluence in the front end of the amplified envelope with respect to excessive amplification at a higher gain factor. For such inhomogeneous envelopes, a theoretical model has been created in our previous research to guide its compensation based on the amplification delay method, realizing a uniform burst profile with a 2 ms duration for a flat-top input [31]. This method could be appropriately applied in burst-mode systems with lower duty cycles, input energies, pump powers, and thermal deposition because of the easier establishment of dynamic equilibrium during pumping and amplification. Nevertheless, the method places restrictions on emerging situations in which there are existing thermal fluences, such as higher pump injection, increased measurement length, as well as lessened periodic thermal release. For this reason, the model was further improved based on the reverse process of amplification. This enhanced the effective compensation for the burst profile on the scale of temporal distribution and gave the model a positive capacity to counteract the gain deterioration induced by thermal effects. In addition, the improved method enables the suppression of pulse distortion due to the gain saturation effect for the purpose of inhibiting optical damage originating from the compressed pulse. This method is proposed to realize synchronous applications in pulse training systems with temporal shaping in both the burst envelope and pulse profile in terms of variable envelope width and stretched pulses at different frequencies.

In this paper, a narrow linewidth, pulse burst laser source was illustrated. The source has an emerging homogenous exhibition in both burst profile and pulse waveform at a 10 kHz rate using a hybrid fiber-bulk amplifier laser system. The resulting irregular deformation realized effective compensation, with a uniform distribution on temporal scales. The coefficient of variation (COV) within a burst was 1.92%, 2.14% and 1.84% over a 4 ms duration. Such a laser system possesses the latent capacity for amplification in optical-based imaging using the FRS diagnosis technique.

## 2. Theoretical Method for Temporal Flat-Top Shaping

Starting from pulsed laser amplification, a theoretical method based on the reverse process of amplification is introduced here for temporal flat-top shaping, where the input is solved for the desired pre-shaped output. The method can beapplied in a burst-mode, solid-state amplifier system over variable envelope widths with long pulse durations.

Frantz and Nodvik et al. presented a basic nonlinear relationship for pulse amplification with the solution of rate equations in a laser amplifier [32]:(1)$$J_{out}=J_{s}\ln\left\{1+G_{i}\left[\exp\left(\frac{J_{in}}{J_{s}}\right)-1\right]\right\}$$
where *G_i_* is the initial gain factor before amplification, *J_s_* is the saturation energy factor for the gain medium, *J_in_* is the input energy density, and *J_out_* is the corresponding output.

The gain distribution after amplification can be characterized using the expression below:(2)$$G\left(t\right)=\frac{G_{i}\exp\left(\frac{pJ_{in}}{J_{s}}\right)}{{\left\{1+G_{i}\left[\exp\left(\frac{J_{in}}{J_{s}}\right)-1\right]\right\}}^{p}}$$
where *p* (0.5 ≤ *p* ≤ 1) is the recovery coefficient used to describe the degree of gain recovery in the case of an equivalent degeneration or no degeneration between the upper level and lower level, with the two critical values of being fully recovered (*p* = 0.5) and not being completely recovered (*p* = 1).

The pre-shaped process is modeled in two steps during double-pass operation. Firstly, the reverse process of amplification is solved by utilizing Equations (1) and (2) repetitively, and the results of Equations (3)–(5) concern the case without complete recovery with *p* = 1. Secondly, an iterative process is utilized to characterize the relevant input distribution corresponding to the temporal uniform output. In this model, some basic assumptions are given, for example, that there are no amplified spontaneous emission (ASE) effects and a negligible pump time during amplification.
(3)$$J_{1,in}=J_{s}\ln\left[1-\frac{1-\exp\left(\frac{J_{1,out}}{J_{s}}\right)}{G_{i}}\right]$$
(4)$$J_{2,in}=J_{s}\ln\left[1-\frac{1-\exp\left(\frac{J_{2,out}}{J_{s}}\right)}{G_{1}}\right]$$
(5)$$G_{1}=\frac{G_{i}\exp\left(\frac{J_{1,in}}{J_{s}}\right)}{1+G_{i}\left[\exp\left(\frac{J_{1,in}}{J_{s}}\right)-1\right]}$$
where *G*_1_ is the gain factor after the first pass; *J*_2,*in*_ is the input of the output *J*_2,*out*_; and *J*_1,*in*_ is the input for output *J*_1,*out*_. The equation relevant to *J*_1,*in*_ is presented in Equation (6) by substituting Equations (4) and (5) into Equation (3) when performing the algebraic operation using *J*_2,*in*_ as *J*_1,*out*_:(6)$$G_{i}^{2}{\left[\exp\left(\frac{J_{1,in}}{J_{s}}\right)\right]}^{2}-\left\{G_{i}^{2}+G_{i}\left[\exp\left(\frac{J_{2,out}}{J_{s}}\right)-1\right]\right\}\exp\left(\frac{J_{1,in}}{J_{s}}\right)-\left(1-G_{i}\right)\left[\exp\left(\frac{J_{2,out}}{J_{s}}\right)-1\right]=0$$

For long pulse amplification, it is noted that the time spanning between entering and leaving the gain medium is much shorter than that of the pulse duration. Hence, the sub-pulse sequence is created through dividing individual pulses into segments, with the relevant distribution presented in Figure 1a. The pulse shape is regarded as consisting of *n* divided sub-pulses, with identical duration *t_p_*/*n* (round trip time within the gain medium) and without any interval between the sub-pulses. Figure 1b illustrates the time sequence distribution in the pulse burst laser involved in a cycle of initial amplification delay *τ*, pulse amplification *t_p_*, storage recovery *t_c_*, and thermal relaxation *t_r_*. Meanwhile, a detailed distribution during *t_p_* is shown in Figure 1a. The stored energy is obtained inside the gain medium at amplification delay *τ* through Equation (3) via the solution of a differential equation [33] with respect to the small signal gain *G*_0_ in Equation (8):(7)$$E_{st}\left(t\right)=\frac{PT\tau_{f}}{S}\left(1-\exp\left(-\frac{t}{\tau_{f}}\right)\right)$$
where *P* is the pump’s peak power, *τ_f_* is the fluorescence lifetime of the gain medium, *T* is the transfer factor of energy storage, and *S* is the effective beam area.
(8)$$G_{0}=\exp\left(\frac{J_{st}}{J_{s}}\right)$$
where small signal gain factor *G*_0_ is related to energy storage density *J_st_* and saturation energy factor *J_s_*.

The pre-shaping of individual pulses is determined starting from the first sub-pulse during the double-pass operation: (i) solve for *J*_1,*in*_ in Equation (6) by replacing initial gain *G_i_* with *G*_0_ at the desired flat-top output; (ii) endow *J*_1,*in*_ in Equation (5) for *G*_1_, substitute *G*_1_ into Equation (4) for *J*_2,*in*_, and another *J*′_1,*in*_ is obtained in Equation (3) utilizing *J*_2,*in*_ as *J*_1,*in*_; and (iii) the iteration process quickly yields convergency by replacing *J*_1,*in*_ with *J*^′^_1,*in*_. Furthermore, the following is completed by utilizing *G*_2_ in Equation (9) (i.e., the gain factor after double-pass amplification) as the initial gain *G_i_* for the next sub-pulse and repeating the above-mentioned steps (i)–(iii). The solved input for the whole pulse is the sum of each segment with respect to the output:(9)$$G_{2}=\frac{G_{1}\exp\left(\frac{J_{1,out}}{J_{s}}\right)}{1+G_{1}\left[\exp\left(\frac{J_{1,out}}{J_{s}}\right)-1\right]}$$

As for the pre-shaping of the burst envelope, it is regarded as combining of a set of individual pulses. Extraordinarily, there is a recovery process during time *t_c_*, resulting in another gain distribution *G*(*t_c_*) in Equation (10), which is utilized as initial gain *G_i_* when solving the input of the next pre-shaped pulse (sub-pulses sequence). In this case, the equation is repeated continuously for the following duration within the envelope, and the basic intensity characterization is given by Equation (11) during the envelope duration:(10)$$G\left(t_{c}\right)=G_{0}^{1-\exp\left(-t_{c}/\tau_{f}\right)}G_{a}^{\exp\left(-t_{c}/\tau_{f}\right)}$$
where *t_c_* is the recovery time for the gain medium and *G_a_* is the gain distribution after the pulse (sub-pulses sequence), which is equal to the gain distribution after the double-pass operation for the last sub-pulse within the main pulse:(11)$$J_{in}\left(t\right)=\int_{0}^{t}I\left(t^{\prime }\right)dt^{\prime }$$

The model is established based on a reverse process of amplification for temporal flat-top shaping in both the pulse profile and burst envelop during the double-pass operation. Additionally, several circumstances ought to be paid attention to for sustaining the model. One is to take into account the beam variation characteristics when utilizing the output as the input for performing the iterative process. The other is that the model makes available different envelope durations at arbitrary frequencies, unless a recovery time exists between pulses within a burst. In this model, the adopted pump source is operated in a pulsed manner.

In addition, this model could extend itself to applications in a multi-stage amplification configuration. The pre-shaped process is regarded as starting with the last stage and combines with each stage utilizing the output as input in the case where the fill factor for each stage is known. Hence, the initial input possesses the solution for the targeted output by an iterative method repeating the steps mentioned above.

## 3. Experimental Setup

Figure 2 exhibits the experimental schematic of a fiber-bulk pulse burst laser system, with two primary segments of both an all-fiber system and a free-space operation. The continuous wave (CW) fiber laser source, over a 1.17 cm^−1^ span of the tunable emission spectrum (1064.109–1064.241 nm), was seeded into an integrated Yb-doped fiber amplifier (YDFA) module with two amplifier stages to scale the laser’s power; the relevant optical configuration of the YDFA is illustrated in [31]. An acousto–optic modulator (AOM), with a high contrast ratio, connects the two fiber ports into the optical path, establishing a train of the burst envelope characterized by the desired intensity distribution. Specifically, the programmable analogue waveform generator (AWG) was added on to the AOM driver; the AWG can emit radio frequency (RF) signals acting on the AOM modulator. Moreover, the established travelling acoustic wave inside the modulator yields the incident light being diffracted. In addition, the amplitude of such diffracted light is directly dependent on the RF signal power, as well as being indirectly subject to the analog electric signal intensity. The resulting burst-mode laser was coupled through a fiber collimator with a focal length of 11.17 mm.

Free-space amplification was arranged in a double-pass amplification layout with three diode-pumped Nd:YAG amplifier modules (0.6 at.% Nd^3+^ doping). First, two amplifier modules with rod diameters of 3 mm (AMP1, AMP2) were connected to each other in series, and another one with a rod diameter of 6 mm (AMP3) was placed in a separate optical path. The sequence of amplifiers for a pulse traveling within a double-pass configuration is AMP2, AMP1, AMP1, AMP2, AMP3, and AMP3. A group of plane-convex lenses (*f*_L1_ = +50 mm, *f*_L2_ = +50 mm) was used to relay excellent seed images for initial amplification characterization. Furthermore, the amplified beam diameter was increased to ~2× by means of a combination of L3 and L4 (*f*_L3_ = −50 mm, *f*_L4_ = +100 mm), with proper matching with AMP3 being realized. An optical isolator (OI), for a peak isolation over 35 dB, was inserted to protect optical elements from being damaged by the reverse feedback beam. To reduce the ASE effect and purify the amplified beam, a spatial filter (SF) was introduced into the double-pass optical path. M1-M6 are presented as 45° reflectors with high-reflectivity (HR) coated at 1064 nm, and M7-M8 served as the 0° reflectors.

In this experiment, the burst envelope and pulse shape were collected with a Si-biased detector (Thorlabs, 200–1100 nm), as well as being recorded with an oscilloscope (KEYSIGHT, DSOX3104T). The amplified pulse energy was measured with an energy detector (Coherent, J-50MT-10KHZ) in combination with an energy meter (Coherent, LabMax-TOP). In terms of the burst-mode system, the interval for the burst envelope was separated by 100 ms to release the thermal deposition accumulated inside the gain medium.

## 4. Results and Discussions

Modulated pulses with different durations were characterized by the energy distribution over the variable envelope width at a 10 kHz pulse frequency, as shown in Figure 3a. The output pulse energy was linearly increased with pulse duration with slope efficiencies of 94.8%, 92.2%, and 95.2%, corresponding to envelope widths of 2 ms, 3 ms, and 4 ms, respectively, due to the growing duty cycle for the injected AWG signal. As a comparison of the same pulse duration, modulated pulses possessed similar energy distribution trends across different envelope widths. Meanwhile, the time-averaged pulse energy increased from 107 nJ at a 100 ns pulse duration to 292 nJ at a 300 ns pulse duration when hte absorbed pump power was 2.43 W below the threshold of the stimulated Brillouin scattering (SBS) effect occurring in the fiber amplifiers. Figure 3b–d illustrate the modulated burst profile distributions over the recorded length of 2 ms, 3 ms, and 4 ms at a 300 ns pulse duration, resulting in COVs within the envelope of 0.69%, 0.81%, and 0.74%, respectively. The weak deviation in the intensity of intra-burst pulses stemmed from periodic oscillations in the AOM’s efficiency determined by independent phases of the AOM RF driver; thus, a stable phase-locked AOM driver could enhance the envelope distribution.

For the modulated flat-top input, the output burst profiles during double-pass operation at amplification delays of 0.2 ms to 0.6 ms over recording lengths of 2 ms, 3 ms, and 4 ms, respectively, are exhibited in Figure 4a–o. First, several pulses within a burst extracted less energy storage at lower amplification delay (0.2 ms and 0.3 ms) and remained greater for the latter during pulse train amplification, yielding a gradually increasing trend in the front end of the burst envelope. Moreover, a decreasing distribution was observed for higher amplification delays (0.5 ms and 0.6 ms) at variable envelope widths due to the reduced gain factor induced by the initial higher energy extraction. However, there was a nearly uniform distribution of subsequent pulses within the envelope over the 2 ms envelope duration, slightly descending for the 3 ms envelope duration, and then significantly decreasing at the 4 ms envelope duration in terms of arbitrary amplification delay. Such emerging differences can be explained by the thermal lens’s effect. The lens’s influence was enhanced by the accumulated heat inside the gain medium during the burst profile with a longer duration, resulting in the deterioration of the distribution when the pulse burst train proceeded. The COV within envelope showed an approximately linear decrease, as plotted in Figure 4p, when starting from both ends of the amplification delay (0.2 ms and 0.6 ms). I also reached its minimum value at 0.42 ms. It can be seen that a nearly flat-top burst profile was realized over 2 ms duration with a COV of 3.01% at a 0.42 ms amplification delay, and restrictions were placed on the establishment of a dynamic equilibrium between pumping and amplification during longer envelope durations due to the thermal impacts. The input of the presented output had an average pulse energy of 292 nJ at a 10 kHz frequency and a 300 ns pulse duration. In addition, the distribution tendency of the burst profile was consistent when varying the input with additional pulse durations in the experiment.

In this work, the Nd:YAG-based modules demonstrated performances endowed with fluorescence lifetime *τ_f_* = 230 μs, saturation energy density *J_s_* = 0.667 *J*/cm^2^, pump peak power *P* = 900 W (AMP1, AMP2), and *P* = 2500 W (AMP3). Losses of 14.5% (AMP1+AMP2) and 10% (AMP3) were measured before the pulses traveled into the amplifiers, and the loses were 39% (AMP1+AMP2) and 17% (AMP3) after one round trip within the optical cavity. Moreover, the effective beam section could be characterized with thermal lensing via the He-Ne source. Figure 5a–c show the input of a pre-shaped burst profile based on the theoretical method within the 4 ms envelope with a 300 ns pulse duration and in terms of amplification delay from 0.42 ms to 0.6 ms. The input distribution exerted a similar trend with the reverse of the trend exhibited above, thus eliminating the inequilibrium distortion occurring in the 4 ms envelope. The flat-top envelope of the pre-shaped output is presented in Figure 5d–f, corresponding to the COV distributions of 1.92%, 2.14%, and 1.84% within a burst. As a comparison between amplification delay methods, our propose method can effectively enhance the burst profile distribution and improve envelope planeness.

In double-pass pulsed amplifiers, the amplified pulse shape was recorded and investigated at different stages with pulse durations of 100 ns to 300 ns over a 4 ms envelope width and a 10 kHz pulse frequency, as presented in Figure 6a–c. For the square seed pulse, it was found that the pulse profile transitioned from being almost unchanged to subtlely decreasing and then obviously dropping after the first stage (AMP1+AMP2) for pulse durations of 100 ns to 300 ns, respectively. The first parts of the pulse could produce a higher gain, and this gain was depleted as the pulse continued traveling within double-pass amplification because the increased input resulted in more particle consumption occuring on the leading edge. Moreover, there is a significant pulse steepening at the leading edge after the final stage (AMP3), resulting in serious pulse shape distortion. In addition, pulse duration showed no obvious change on the whole. Considering such distortion, the main purpose of the shaping capacity is to counteract the effects of gain saturation in the solid-state amplification chain. The seed profile was solved so that the front end of the pulse profile possessed less energy than that of the latter, thus compensating for gain depletion. The input and output of the pre-shaped flat-top pulse profile is exhibited in Figure 6d–f for pulse durations from 100 ns to 300 ns, indicating a positive pulse shaping capacity. The practical AWG signal of the AOM contained a 2.5 W RF driving power, the amplitude of which could be controlled.

In order to enhance the amplified pulse energy, burst-mode operation was performed with a 300 ns pulse duration at a 0.42 ms amplification delay, as illustrated in Figure 7a. The amplified pulse energy significantly increased with the total injected pump peak power. The maximum pulse energy scaled up to 9.68 mJ, 8.94 mJ, and 6.57 mJ at a pump peak power of 4300 W with respect to the pre-shaped output. As a longitudinal comparison, the enlarged pulse energies for the 2 ms and 3 ms envelope durations possessed similar distributions until separating at the nearly maximum pump peak power. Furthermore, such a trend showed a breakdown as the envelope width expanded to 4 ms due to serious gain deterioration. The inset demonstrates the variation in the maximum pulse energy versus the amplification delay for different envelope durations. Correspondingly, there was no obvious output fluctuation at 2 ms and 3 ms envelope durations when increasing the amplification delay to 0.7 ms. However, a faster drop occurred for the 4 ms burst profile, which could reflect the model’s susceptibility to thermal imbalance from the side to some extent over longer envelope durations. Figure 7b presents the operation stability of the pulse burst laser with the maximum output in 10 min. The standard deviations of the system’s operation were 2.09%, 2.67%, and 2.48% in terms of the 2 ms, 3 ms, and 4 ms burst durations, respectively, indicating the relatively stable operation of our burst-mode laser system.

The output spectral profile of the pre-shaped beam was measured by utilizing a scanning Fabry–Perot interferometer at full pump energy over a 4 ms burst envelope and a variable pulse duration. From Figure 8a–c, the measured spectral bandwidth, FWHM of Lorentz fit was 68.3 MHz, 67.2 MHz, and 67.7 MHz, corresponding to pulse durations of 100 ns, 200 ns, and 300 ns, respectively. Also shown is the convolution of the Fabry–Perot cavity mode and Fourier transform of practical temporal beam of the MOPA output. In addition, a longer pulse could result in narrower bandwidths, as defined by Heisenberg’s uncertainty principle [34]. For this work, the measured spectral linewidth approached the resolution (67 MHz) of the Fabry–Perot interferometer with a free spectral range (FRS) of 10 GHz. Hence, a precise interferometer with higher resolution could better distinguish the spectral profile for more accurate spectral linewidth characterization.

## 5. Conclusions

To sum up, a 10 kHz narrow linewidth burst-mode source with a pre-shaped temporal profile was investigated via a hybrid fiber-bulk master oscillator power amplifier (MOPA) system. The theory of the relevant method was illustrated and proposed for burst-mode shaping by solving for the input of the desired pre-shaped output based on a reverse process of amplification. Furthermore, such a theoretical model was created by means of a successively iterative process. Modulated seedswere characterized by an energy distribution increasing from 107 nJ to 292 nJ in terms of a pulse duration ranging from 100 ns to 300 ns and an absorbed pump power of 2.43 W. As for the flat-top input, othe utput profile was ultimately recorded during a double-pass operation at variable envelope widths, pulse durations, and amplification delays. Limitations were also introduced to establish a dynamic equilibrium between the pumping and amplification processes during longer envelope durations to address thermal impacts. Therefore, the solved input, based on the theoretical method, effectively compensated for the inequilibrium distribution within a burst, yielding COV distributions of 1.92%, 2.14%, and 1.84% for envelope durations of 2 ms, 3 ms, and 4 ms, respectively. In addition, the injection of a pre-shaped pulse profile possessed the positive capacity to combat the excessive depletion of particles occuring on the leading edge in a solid-state amplification chain. The amplified pulse energy was as high as 9.68 mJ, 8.94 mJ, and 6.57 mJ over envelope widths of 2 ms, 3 ms and, 4 ms, respectively, and a 300 ns pulse duration, corresponding to standard deviations of burst-mode systems of 2.09%, 2.67%, and 2.48%. The spectral linewidths were expressed as 68.3 MHz, 67.2 MHz, and 67.7 MHz in terms of pulse durations ranging from 100 ns to 300 ns. Compared with previous amplification delay methods, this method efficiently counteracted thermally induced envelope malformations and pulse distortions caused by front-end gain saturation.

## Figures and Tables

**Figure 1 sensors-23-00834-f001:**
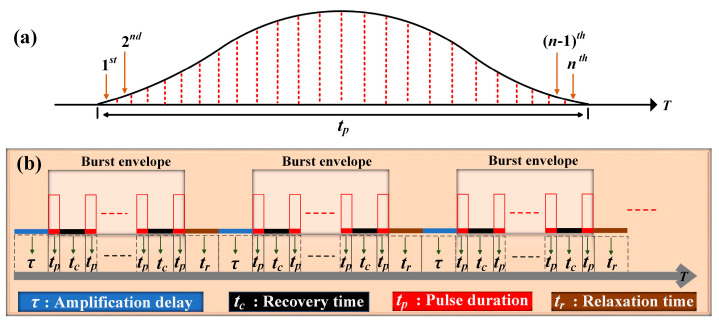
(**a**) Distribution of divided sub−pulses sequence for intra−burst individual pulse; (**b**) Schematic of time series distribution in burst-mode laser system.

**Figure 2 sensors-23-00834-f002:**
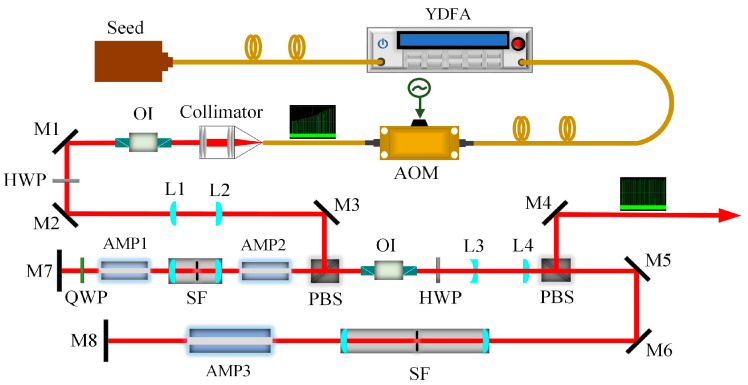
Experimental schematic of the pulse burst MOPA laser system. YDFA: Yb-doped fiber amplifier; AOM: acoustic–optic modulator; HWP: half-wave plate; QWP: quarter-wave plate; PBS: polarizing beam splitter; OI: optical isolator; SF: spatial filter.

**Figure 3 sensors-23-00834-f003:**
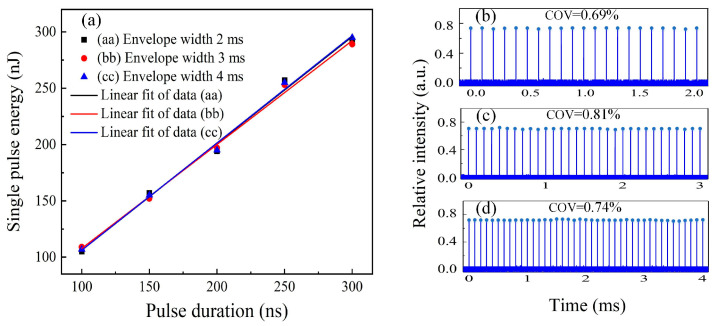
(**a**) Output pulse energy versus pulse duration over envelope widths of 2 ms to 4 ms at 10 kHz frequency; (**b**) 2 ms burst profile distribution; (**c**) 3 ms burst profile distribution; (**d**) 4 ms burst profile distribution.

**Figure 4 sensors-23-00834-f004:**
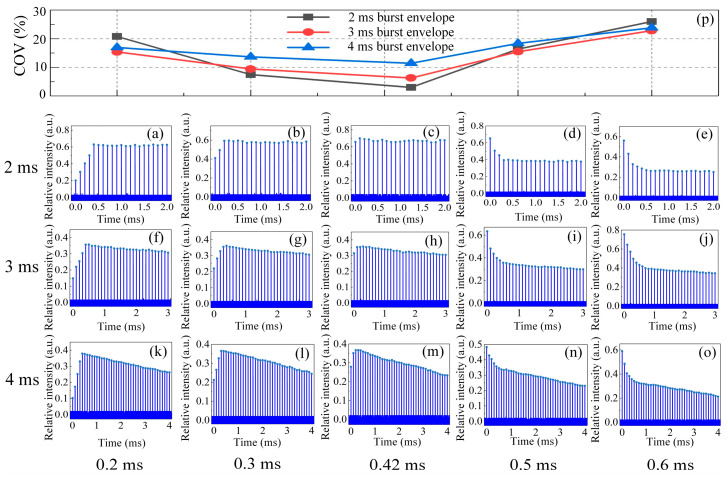
Recorded output burst profile and COV distribution for uniform input at amplification delays of 0.2 ms to 0.6 ms over variable envelope durations: (**a**–**e**) 2 ms burst profile; (**f**–**j**) 3 ms burst profile; (**k**–**o**) 4 ms burst profile; (**p**) COV distribution versus amplification delay at different envelope durations.

**Figure 5 sensors-23-00834-f005:**
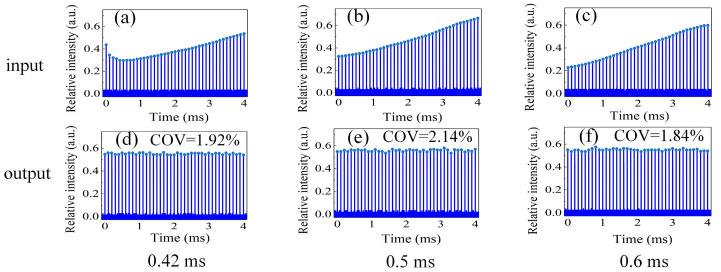
Input and output envelope of pre-shaped flat-top burst profile at amplification delays of 0.42 ms, 0.5 ms, and 0.6 ms. (**a**–**c**): Input profile of pre-shaped envelope; (**d**–**f**): Output profile of pre-shaped envelope.

**Figure 6 sensors-23-00834-f006:**
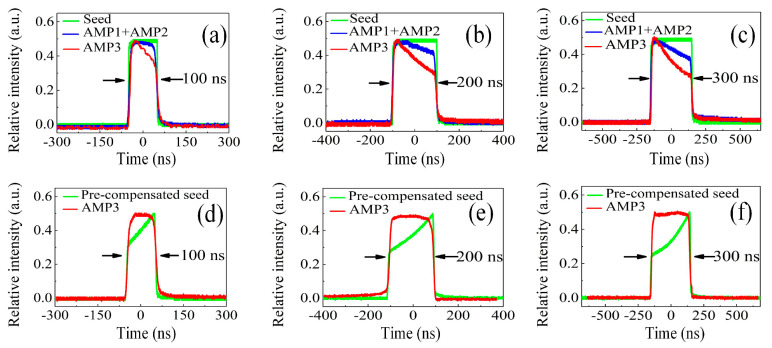
Recorded input and output pulse profile for pulse durations of 100 ns, 200 ns, and 300 ns. (**a**–**c**): Output profile of flat-top input at different amplification stages; (**d**–**f**): Input and output of pre-shaped flat-top profile.

**Figure 7 sensors-23-00834-f007:**
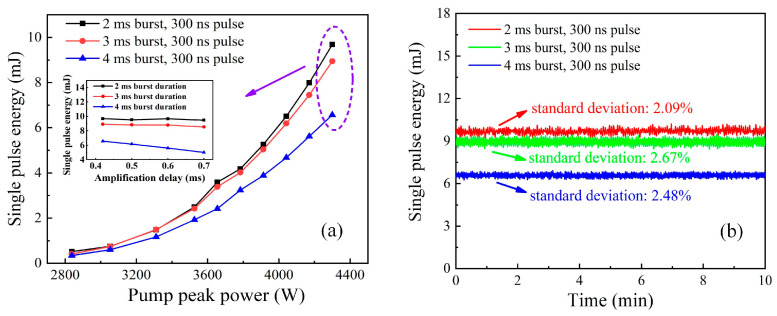
(**a**) Output pulse energy versus input pump peak power at amplification delay of 0.42 ms (Inset: amplified delay versus pulse energy for flat-top envelope and pulse profile at full pump energy); (**b**) Corresponding operation stability of pulse burst MOPA laser system for uniform envelope distribution at flat-top pulse profile.

**Figure 8 sensors-23-00834-f008:**
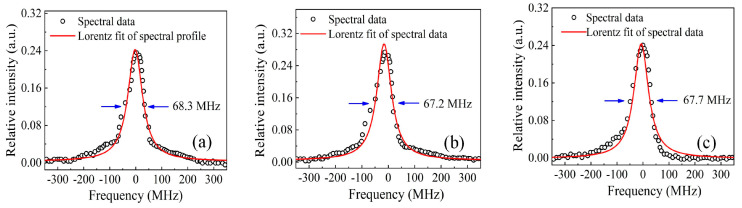
Time-averaged spectral profile of pulse burst MOPA laser for variable pulse width over 4 ms uniform envelope: (**a**) At 100 ns pulse duration; (**b**) At 200 ns pulse duration; (**c**) At 300 ns pulse duration.

## Data Availability

The data presented in this study are available on request from the corresponding author.

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
