# Peer review of "Pre-Shaped Burst-Mode Hybrid MOPA Laser System at 10 kHz Pulse Frequency"

_sensors, 2023, doi:10.3390/s23020834_

Round 1

Reviewer 1 Report

The authors reported an improved theoretical method based on reverse process of amplification used for promoting burst envelope and pulse profile shaping in pulse burst laser system simultaneously. By contrast with so-called “amplification delay method”, it embodied flexibility and diversity. In my opinion, this paper holds a certain degree of innovation, in spite of lower energy index. This manuscript could fulfill a request for publication on Sensors after finishing with some revision. The addressed issues are listed as below

1. Concrete information ought to be added in this paper such as adopted instrument model and crystal sizes.

2. Whether authors could explain to me the so-called “amplification delay method”? How does it achieve flat-top distribution? I am not clear to it.

3. Authors mentioned that thermal accumulation could lead to inhomogeneous distribution in burst-mode laser. Why does this happen? Please explain it.

Author Response

Dear Editor and Reviewer

We are grateful for the suggestions and comments with respect to our manuscript by the reviewer. Please find below our responses in “red fonts” to the reviewer’s questions and comments.

We attach a revised manuscript based on the reviewer’s suggestions where our revisions are indicated in “red fonts”. 

Please contact us, if you should have further questions. 

With best regards,

Authors

1.Concrete information ought to be added in this paper such as adopted instrument model and crystal sizes?

Thanks for reviewer’s suggestion. In the experiment, the burst envelope and pulse shape were collected with a Si-biased detector (Thorlabs, 200-1100 nm,) as well as recorded with oscilloscope (KEYSIGHT, DSOX3104T). Amplified pulse energy was characterized with energy detector (Coherent, J-50MT-10KHZ), in combination of energy meter (Coherent, LabMax-TOP). Concrete information of adopted instrument model have been added in this manuscript on page 7. Meantime, AMP1 and AMP2 were provided with rod diameter of 3 mm with respect to 6 mm rod diameter for AMP3, the relevant information has been added in this paper on page 6. Please check it.

2. Whether authors could explain to me the so-called “amplification delay method”? How does it achieve flat-top distribution? I am not clear to it.

Thanks for reviewer’s suggestion. For flat-top input, initial energy storage could induce inhomogeneous distribution within burst envelope. The “amplification delay method” could compensate for burst profile by means of optimizing amplification delay, under the action of theoretical model. The flat-top burst envelope was generated in the end, with the establishment of dynamic equilibrium in pumping and amplification.  

3. Authors mentioned that thermal accumulation could lead to inhomogeneous distribution in burst-mode laser. Why does this happen? Please explain it.

Thanks for reviewer’s suggestion. Inhomogeneous distribution in burst-mode laser could give its explanation with thermal accumulation. Its influence was enhanced by accumulated heat inside gain medium during burst profile with long duration due to thermal-induced transient focal length. Such factors could result in uneven energy extraction, and then deterioration of the distribution when pulse burst train proceeded. The relevant information has been added on page 8. Please check it.   

Reviewer 2 Report

In this manuscript, the authors demonstrate a temporal pre-shaped burst-mode hybrid fiber-bulk laser system with emerging homogenous exhibition in both burst profile and pulse waveform. A theoretical model was proposed and abundant experimental work was implemented. The claimed performance is very good. However, I have some comments and questions which need to be clarified before publication.

1. In introduction, please keep line spacing consistent across paragraphs.

2. In Eq. (1), Jin and Jout should also be defined.

3. Usually, we define pulse duration full width at half maximum of the pulse envelope, but in Fig. 1(a), the pulse duration covers the entire pulse envelope. What’s the difference?

4. In line 209-210, the statement regarding the equality of AMP1 and AMP2 does not correspond to the statement in Figure 2. 

5. In Fig. 7 (b), how does the author measure the operating stability of the pulsed pulsed laser at its maximum output over a continuous period of 10 minutes? 

6. The T here is the text distortion due to compression in Figures 3, 4, and 5. 

Author Response

Dear Editor and Reviewer

We are grateful for the suggestions and comments with respect to our manuscript by the reviewer. Please find below our responses in “red fonts” to the reviewer’s questions and comments.

We attach a revised manuscript based on the reviewer’s suggestions where our revisions are indicated in “red fonts”. 

Please contact us, if you should have further questions. 

With best regards,

Authors

1. In introduction, please keep line spacing consistent across paragraphs.

Thanks for reviewer’s suggestion. The line spacing across paragraphs has been kept in consistence in introduction.  

2. In Eq. (1), Jin and Jout should also be defined.

 Thanks for reviewer’s suggestion. Jin is the input energy density, with respect to output energy density Jout. The relevant information has been added on page 3 in the revised manuscript.  

3. Usually, we define pulse duration full width at half maximum of the pulse envelope, but in Fig. 1(a), the pulse duration covers the entire pulse envelope. What’s the difference? 

Thanks for reviewer’s suggestion. Full width at half maximum of the pulse envelope was defined as pulse duration. Differently, individual pulse in Fig. 1(a), covering the entire pulse envelope, was divided into plenty of sub-pulses with the same round-trip time within gain medium. Such division could effectively reflect pulse distortion due to gain saturation occurring on leading edge during flat-top pulse amplification.   

 4.  In line 209-210, the statement regarding the equality of AMP1 and AMP2 does not correspond to the statement in Figure 2.  

Thanks for reviewer’s suggestion. Sequence of amplifiers for a pulse traveling within double-pass configuration has been modified as AMP2, AMP1, AMP1, AMP2, AMP3, and AMP3. The relevant information has been corrected on page 6 in the revised manuscript. Please check it.  

5.  In Fig. 7 (b), how does the author measure the operating stability of the pulsed laser at its maximum output over a continuous period of 10 minutes? 

 Thanks for reviewer’s suggestion. The adopted energy detector was provided with 10 kHz maximum operation rate. Correspondingly, energy meter was set with record time of 600 s, as well as with averaged pulse energy per second within collected arbitrary 40 pulses. In this case, the operating stability of the pulsed laser was measured at its maximum output over a period of 10 minutes.  6. The T here is the text distortion due to compression in Figures 3, 4, and 5.  

Thanks for reviewer’s suggestion. The compressed text distortion has been modified in Figures 3, 4, and 5 respectively. The relevant information has been corrected on page 7, 8, and 9 in the revised manuscript. Please check it.  
